Journal of Data-centric Machine Learning Research (2026)    Submitted 4/26; Revised xx/xx; Published xx/xx

# The Unreasonable Benchmark

**Fernando Perez-Cruz**[1,2]*,    **Briland Hitaj**[3]*,    **Giovanna Maria Dimitri**[4]*,    **Oishi Deb**[5]*, **Amine M'Charrak**[5], **Shikhar Srivastava**[6], **Christopher Kanan**[6], **Anup Kumar Gupta**[7], **Zefang Liu**[8], **Zihan Zhu**[9], **Tarun Kumar**[10], **Álvaro Galisteo Bermúdez**[11], **Vasilios Mavroudis**[13], **Yuyang Xue**[14], **Brian Matejek**[3], **Swati Rajwal**[15], **Avinash Kumar Pandey**[15], **Ayyüce Begüm Bektaş**[16], **Anil B Murthy**[18], **Feng Chen**[14], **Hamed Babaei Giglou**[20], **Jennifer D'Souza**[20], **Huascar Sanchez**[3], **Michael Kraus**[17], **Isamu Lautenschläger**[17], **Ziyang Wang**[12], **Xuanli He**[19], **Hyun Song Shin**[1]

[1]*Bank for International Settlements, Basel, Switzerland,* [2]*Computer Science Department, ETH Zürich, Switzerland,* [3]*Computer Science Laboratory, SRI International, Menlo Park, CA 94025, USA,* [4]*University of Milan, Milan (Italy),* [5]*University of Oxford, UK,* [6]*University of Rochester, USA,* [7]*Indian Institute of Technology Indore,* [8]*Georgia Institute of Technology, USA,* [9]*School of Computing, FSE, Macquarie University, Australia,* [10]*Hewlett Packard Labs, USA,* [11]*University of Seville, Spain,* [12]*Aston University, Birmingham, UK,* [13]*King's College London, UK,* [14]*The University of Edinburgh, UK,* [15]*Emory University, USA,* [16]*Sloan Kettering Institute for Cancer Research, USA,* [17]*TU Darmstadt,* [18]*School of Computing and AI, Arizona State University, USA,* [19]*University College London, UK,* [20]*TIB Leibniz Information Centre for Science and Technology*

**Reviewed on OpenReview:** *https: // openreview. net/ forum? id= XXXX*

**Editor:** Lijie Hu

## Abstract

Effective benchmarking is instrumental for the systematic evaluation of Large Language Models (LLMs) performance, providing vital insights in the path towards advancements in Artificial Intelligence (AI). Existing benchmarks, such as Humanity's Last Exam and FrontierMath, stress-test the boundaries of the model's performance. However, they provide limited understanding of a model's reliability in routine, low-difficulty scenarios. It is, however, these contexts, which, despite their simplicity, frequently expose persistent model weaknesses. To address this, we present the `Dataset`, a complementary benchmark aiming to address this oversight via systematic (and continuous) evaluation of LLMs' performance in basic reasoning and everyday tasks. It is the end result of a crowdsourcing effort seeking to ensure a diversity of perspectives and topic coverage. Moreover, the dataset is designed to be *dynamic* in nature; it incorporates new items as emerging failure modes are identified while retiring resolved items, thereby maintaining relevance over time. Similar to previous efforts, such as TruthfulQA, the primary objective of this new benchmark is to identify persistent shortcomings and support the development of robust and reliable LLMs. Ultimately, this process should converge upon a set of *unreasonable* issues, at which point we can be confident that the LLMs will indeed exhibit impressive and resilient

---

*. Joint First Authorship (the order is random) and Corresponding Authors.
  Contact: `fernando.perez-cruz@bis.org`; `briland.hitaj@sri.com`; `giovanna.dimitri@unimi.it`; `oishideb@robots.ox.ac.uk`

capabilities. The set of (multimodal) questions is available `https://huggingface.co/datasets/unreasonablebenchmark/unreasonable-benchmark`, under a Creative Commons Attribution-ShareAlike (CC BY-SA) license.

**Keywords:** benchmarks ; LLMs ; unreasonable benchmarks

## 1 Introduction

Murphy (2012) defines the core objective of machine learning (ML) as being the development of models that make accurate predictions. In this vein, ML benchmarking has therefore been and still remains, an essential mechanism for assessing progress. Benchmarking ensures that progress is not evaluated merely by surpassing prior methods; benchmarking has been serving as the basis for determining when substantive advances have occurred and even deciding when a problem (or set of problems) can be deemed resolved.

The advent of large language models (LLMs), and particularly the zero-shot capabilities that accompany them, has reshaped both the benchmark design and the interpretation of the model performance. LLM's generality challenges task-specific evaluation paradigms, making a lot of benchmarks obsolete or limited in scope. Indeed, even broad assessment benchmarks – such as the Humanity's Last Exam, see Phan et al. (2025) – provide coverage only for a subset of relevant capabilities, and more generally, benchmark claims depend on whether the underlying tasks and metrics validly capture the target capability Bean et al. (2025).

As benchmarks for LLMs become entrenched, model development constantly optimizes for them, thus reducing their effectiveness as indicators of *genuine* predictive capability. This is in line with Goodhart's Law: "Any observed statistical regularity will tend to collapse once pressure is placed upon it for control purposes", see Goodhart (1981). This principle can also be paraphrased as: "*When a measure becomes a target, it ceases to be a good measure*". In the context of LLMs, this translates into a need for benchmarks that evolve to preserve their relevance and diagnostic utility Marro et al. (2026).

This paper takes a first step in this direction. We introduce `Unreasonable Benchmark`[1], a crowdsourced dataset composed of questions of varying difficulty, seeking to shed light on areas and tasks where state-of-the-art LLMs reliably underachieve. Unlike benchmarks designed to surface unexpected strengths, the `Unreasonable Benchmark` dataset focuses on **negative surprises** – instances where models fall short of reasonable performance, despite expectations of success. By documenting such failure modes, the `Unreasonable Benchmark` offers diagnostic value, supporting efforts to enhance model robustness and address systematic errors.

**Why is this relevant?** Artificial General Intelligence (AGI) is often defined as the ability to perform *any* cognitive task a human can do, see e.g., Altman (2024); Morris et al. (2024).This characterization is typically used as a high-level reference point for discussing broad generalization abilities in artificial systems. In this work, however, we do not evaluate or make claims about AGI. Instead, we focus on the more specific question of how well models generalize across different types of tasks, including both challenging and routine ones. Robust performance across diverse task distributions is a key property of reliable machine

---

1. `https://huggingface.co/datasets/unreasonablebenchmark/unreasonable-benchmark`

learning systems, and it is often assumed that such robustness should manifest consistently rather than being limited to narrow subsets of problems. The relationship between performance on controlled benchmark tasks and behavior in real-world applications remains complex and not fully understood. While benchmark evaluations can provide useful signals about model capabilities under standardized conditions, they do not directly translate into guarantees of performance in operational settings. Real-world domains typically involve additional sources of complexity, including variability in data quality, shifting environments, and dependencies on system-level integration and human oversight. Domains such as healthcare and cybersecurity are often cited in discussions of reliability due to their sensitivity to errors and their dynamic, high-stakes nature. In healthcare, for instance, decision support systems must operate under conditions where data may be incomplete or evolving, and where incorrect outputs can have serious consequences. Similarly, cybersecurity environments are characterized by adaptive adversaries and rapidly changing attack strategies, where system effectiveness depends not only on algorithmic performance but also on robust infrastructure, monitoring, and response mechanisms. These examples are used here solely to motivate the importance of robustness and generalization in applied settings, rather than to suggest a direct correspondence with benchmark performance.

The `Unreasonable Benchmark` provides a way into evaluating models on this dimension. The dataset is composed of more than **100**-questions of varying difficulty, spanning not only text-based challenges but also text and image modalities. Furthermore, the dataset is aimed at being *dynamic*, i.e., once an LLM is capable of solving a particular question, that question is removed (marked as "solved"), with new failure cases being added over time. The dataset's utility diminishes only when the remaining questions are no longer seen as reasonable – hence its name. Since we first started asking the community for contributions and collecting questions, only 21 questions have been solved, while others still remain an open problem even for recent state-of-the-art models.

The remainder of the paper is structured as follows: Section 2 reviews existing benchmarks for evaluating LLMs, noting key limitations and motivating the need for a revised approach. Section 3 introduces the proposed benchmark, explains its design rationale, and describes the current question categories. Section 4 presents representative examples of questions for each category while also discussing persistent model failures. In Section 5 we discuss the limits of the current benchmark, whereas Section 6 discusses the ethical considerations revolving this work. We conclude in Section 7 with a summary of contributions and directions for future research.

## 2 Related Work

Several benchmarks have been developed to measure progress in LLM performance. In this section, we review both well-established benchmarks and those closely related to the approach proposed in this paper.

LM Arena by Chiang et al. (2024) is an open, community-driven platform for evaluating LLMs through head-to-head comparisons. Users assess anonymised model responses to the same prompt, contributing to an Elo-based leaderboard that reflects user preferences. While widely adopted, concerns have been raised by Singh et al. (2025), regarding potential bias

toward major AI labs, raising questions related to transparency, fairness, and manipulation susceptibility.

Furthermore, in their work *On the Measure of Intelligence*, Chollet (2019) argues that intelligence lies not in the task-specific performance, but in the efficiency of acquiring new skills across diverse, unfamiliar problems – formalized as "skill-acquisition efficiency" under Algorithmic Information Theory. The author criticized standard AI benchmarks for rewarding scale or hand-coded priors rather than genuine generalization, and outlined principles for fairer evaluation. To implement this vision, he introduces the Abstraction and Reasoning Corpus (`ARC`), a benchmark of abstract, human-solvable, but machine-hard puzzles designed to test reasoning under minimal data and clearly stated priors. This work underpins the ARC Prize, a public $1 million challenge to exceed 85% accuracy on the updated ARC-AGI benchmark, aimed at advancing general-purpose AI. Recently, a fine-tuned version of OpenAI's o3 model reportedly achieved performance at the ARC Prize threshold; yet, it is not regarded as AGI, underscoring the limitations of relying on a single benchmark as a proxy for general intelligence, see Chollet (2024).

In this vein, FrontierMath and Humanity's Last Exam (HLE) are additional examples of advanced benchmarks aimed at testing LLMs beyond standard evaluations. FrontierMath by Besiroglu et al. (2024), is comprised of hundreds of unpublished, research-level problems across disciplines such as number theory, algebraic geometry, and set theory—many requiring significant effort from human experts. Whereas, HLE by Srivastava et al. (2023) is a multimodal benchmark featuring 2,500 expert-curated questions from subjects including mathematics, physics, biology, humanities, and computer science. Designed to be unambiguous and resistant to memorisation, HLE provides a stringent test of model understanding.

On top of the previous examples, several studies have introduced datasets to evaluate model performance on ambiguous questions, with a focus on the inherent human-generated language ambiguity. For example, Min et al. (2020), in a pre-LLM era study, observed that open-domain QA systems often assume a single correct answer, despite the fact that many user questions admit multiple plausible interpretations. Their work was among the first to frame ambiguity as a primary challenge, offering both a dedicated dataset and baseline results to compare with.

In more recent studies, the limitations of large language models (LLMs) in handling ambiguity are highlighted. For instance, Keluskar et al. (2024) demonstrates that state-of-the-art LLMs often misinterpret subtly ambiguous queries, though simple, inference-time prompts can significantly improve accuracy. Whereas Kim et al. (2024) proposes a fine-tuning approach that uses models' self-assessed uncertainty to guide supervision, enabling them to detect and clarify ambiguous input, similar to how reasoning models work today. Li et al. (2025), on the other hand, introduces `CondAmbigQA`, a benchmark that formalizes ambiguity as a structured variable to evaluate system performance across alternative interpretations. Saparina and Lapata (2025) extend these ideas to text-to-SQL generation by incorporating clarification steps into the pipeline.

Even if related, existing work primarily addresses ambiguity arising from human phrasing, rather than probing the inherent limitations of language models. These studies focus on strategies to disambiguate inputs—whether through prompting, fine-tuning, or benchmarking—to help LLMs obtain the correct answers. In contrast, our proposed `Unreasonable Benchmark` dataset, targets a different objective: it highlights cases where questions appear

solvable, yet current models systematically elicit incorrect responses, in the process exposing core capability gaps rather than remediable misunderstanding.

To the best of our knowledge, only eight recent studies target a similar space, presenting questions that current LLMs should, in principle, be able to solve, yet consistently fail to do so—underscoring persistent limitations. Perez-Cruz and Shin (2024) show that non-reasoning LLMs can correctly answer the Cheryl's birthday problem in its original form (`https://tinyurl.com/yh8663va`), yet fail when surface-level changes are introduced—indicating reliance on memorised patterns rather than true reasoning. In contrast, current reasoning models, most likely exposed to the problem during training, correctly solve both original and altered versions identifying the underlying logical structure.

Duchnowski et al. (2025) introduce a benchmark recasting three NP-hard problems—Graph Coloring, Knapsack, and Traveling Salesperson—into eight natural-language variants, including standard formulations, everyday analogues, and inverted forms. Through the evaluation of GPT-4o and LLaMA 3.1 (70B) across multiple prompting strategies, they find that models perform well on textbook variants but struggle with rephrasing, rarely outperforming simple heuristics. Same performance gaps persist even at fixed problem sizes, indicating reliance on memorised patterns over genuine combinatorial reasoning. Further evaluation with reasoning-augmented models is needed to assess whether these limitations remain.

Huckle and Williams (2025) propose a benchmark comprising of puzzles, spatial and relational reasoning tasks, counting problems, linguistic manipulations, and popular-science queries—tasks easily solved by adults, but frequently mishandled by non-reasoning LLMs. They propose a lightweight two-step prompting strategy, requiring models to generate clarifying questions before answering, and obtaining a 41% average performance gain, though results remain well below human baselines.

Mirzadeh et al. (2025) present GSM-Symbolic, a programmatically generated extension of GSM8K that produces controlled families of 8-grade-school math problems, mitigating static-set leakage and enabling systematic perturbation studies. Considering 25 leading LLMs, accuracy under eight-shot chain-of-thought prompting varies widely, with performance sensitive to minor numeric changes. All models, including early reasoning versions (e.g., GPT-4 o1-preview), underperform relative to GSM8K. Accuracy declines with logical complexity, and inserting a single syntactically valid but irrelevant clause can reduce success rates by up to 65%, highlighting the models' reliance on brittle pattern-matching rather than robust symbolic reasoning.

In their work, Giglou et al. (2023) present a framework for LLMs to automate Ontology Learning (OL). The authors evaluate 9 different LLMs using zero-shot prompting across three primary OL tasks. Their findings suggest that LLMs may not be entirely suitable for constructing complex ontologies that require in-depth reasoning and domain-specific expertise.

Li et al. (2024) introduce FLUB, a benchmark designed to assess LLMs ability to comprehend and reason through deceptive or fallacious texts, easily understood by humans, but challenging for machines. FLUB comprises 834 samples sourced from a Chinese online forum known for humorous and misleading content, categorized into 8 types of fallacies. It includes 3 tasks: answer selection, fallacy type classification, and fallacy explanation. The study reveals that current LLMs struggle with these tasks, showing a significant gap be-

tween human and machine understanding of nuanced language, and highlights the need for improved training methods to enhance LLMs' reasoning capabilities.

Bang et al. (2025) propose a comprehensive framework for evaluating hallucinations in LLMs, distinguishing between intrinsic hallucinations (inconsistencies with the input context) and extrinsic hallucinations (content not grounded in the training data). To address the lack of standardised benchmarks, the authors propose a taxonomy and introduce 3 new extrinsic evaluation tasks—PreciseWikiQA, LongWiki, and NonExistentRefusal—that dynamically generate test sets to prevent data leakage and ensure robust assessment.

Last, but not least, TruthfulQA by Lin et al. (2021) is an 817-question benchmark spanning 38 everyday domains (health, law, finance, politics, etc.) that deliberately recite *imitative falsehoods* – answers that people and language models commonly produce because they echo widespread misconceptions rather than facts. The paper further shows that a simple automated metric correlates closely with human judgments arguing, that improving truthfulness will require objectives beyond next-token imitation, such as fine-tuning on human-labeled truthfulness rather than merely scaling model size.

## 3 The `Unreasonable Benchmark` Dataset

TruthfulQA was published in 2022 Lin et al. (2021), prior to the release of ChatGPT, in which the authors highlight the necessity of aligning LLMs to prevent them from reproducing falsehoods embedded in data sourced from the internet. In addition to presenting hundreds of questions that demonstrate how LLMs echo human misconceptions, the significance of this work lies in its emphasis on addressing persistent limitations in LLM performance. It further demonstrated that merely increasing data quantity and model size would be insufficient to overcome these challenges.

The `Unreasonable Benchmark` dataset builds upon this premise by identifying a set of issues that contemporary LLMs fail to manage effectively, and by offering examples to guide developers towards areas where targeted improvement is required. More broadly, this benchmark draws attention to common queries where model responses remain unreliable—issues which, once addressed, could enhance trust in the models' ability to produce robust and accurate outputs in everyday tasks.

We constructed a benchmark of 100 questions contributed by more than 30 individuals to identify cases in which a question was unambiguous to humans yet remained difficult for current LLMs to answer correctly. The benchmark was crowdsourced to improve coverage across perspectives, interests, and lived experiences. To make this diversity more explicit, we included additional evidence on the contributors' geographical distribution, backgrounds, and traditions, details shown in Figure 1.

The version presented in this paper is a snapshot of the unreasonable benchmark as of the submission date in April 2026, with all of the tested LLMs listed in Section 4.5. For evaluation, each question was presented once to each tested LLM in a zero-shot setting (Section 4.4), with no follow-up interaction or iterative prompt refinement. The prompts and corresponding model outputs were then reviewed by human annotators, specifically the original contributors and the first authors of the paper. Concerning scoring protocols,all of the questions reported in the benchmark produced 0% success rate for the prompts fed into the relative LLMs.Answer-validation processes happened through the validation of both submit-

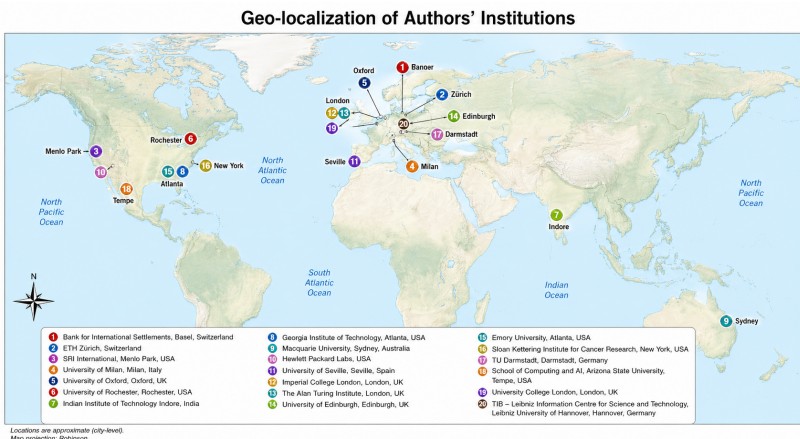

Figure 1: Map showing the distribution of the institutional localization of the authors of the unreasonable benchmark paper

ters and first-authors of the paper. The prompt for each challenge (including accompanying images, where needed) are available with the dataset (and released in the Unreasonable Benchmark repository). We did not apply any modifications to the original prompt (provided by the respective contributors) during the *internal* evaluation process, i.e., our scoring protocol sought to faithfully replicate the same testing conditions provided by the original contributors. Challenges (questions) that are solved by at least one LLM are marked as "solved" and removed from evaluation, with the understanding that subsequent models will soon be able to solve the same question.[2]

Concerning update-rules: the database is intended and designed in such a way to remain dynamic, with new issues added over time and resolved ones removed. In contrast to other benchmarks that aim to remain *pure* by discouraging developers from using them for training, our intention is for this dataset to serve as a source of inspiration for addressing persistent problems. The benchmark will have achieved its purpose once all listed issues are no longer reasonable tests, when no LLM developer considers it a reflection of outstanding challenges.

### 3.1 Dataset Availability

The `Unreasonable Benchmark` dataset has been made publicly available for download and use under a Creative Commons Attribution-ShareAlike (CC BY-SA) license at `https://huggingface.co/datasets/unreasonablebenchmark/unreasonable-benchmark`, under a Creative Commons Attribution-ShareAlike (CC BY-SA) license. This is in line with prior work and will ensure that the dataset continues to grow and easily be used by other researchers in the field. The dataset is composed of a main file served in the `jsonl` format. In addition to this, currently the images are served as a separate compressed (`.tar.gz`) folder.

---

2. The `Unreasonable Benchmark` dataset has a separate field, namely "`is_solved`" to depict questions that are addressed already and should be removed from evaluation. Such questions are kept to capture the evolution of the dataset and transparency. As mentioned earlier in the paper, we anticipate that as the models become stronger (i.e., more intelligent) the number of unsolved questions remaining will decrease.

## 3.2 Data Categories

In Section 4, we present a series of illustrative examples of questions that current models are unable to solve. We bring to the attention of the reader that in the current form, the `Unreasonable Benchmark` dataset is primarily composed of 3-main categories, namely: 1) Text-based, 2) Math-based, and 3) Logical Puzzles. These series of questions test different capabilities of the target LLMs with the data primarily being fed in either *text* or *text + image* modalities. More specifically for text modality we intend a prompt made only of textual inputs. For our benchmark we present only prompts in english. For what concerns instead the *text+image*, the survey form had a specific further field, in which we were asking the authors to further submit the relative image if needed. The only allowed files were image type files (png, jpg or svg) and no pdf or any other sorts of visual files). Since the dataset is designed to serve as a dynamically evolving benchmark, we are planning as future works to include also other data categories and modalities being incorporated as the dataset continues to evolve and grow.

## 4 Example Questions per Category and LLMs Evaluated

In this section, we move on to delve deeper into the current data categories present in the `Unreasonable Benchmark`. For each category, we randomly selected one of the questions, serving as the representative for similar questions currently in the dataset.

### 4.1 Category 1: Text-based Question

As indicated earlier, Section 3, these types of questions were purely presented as text-based input prompts without any other modalities involved. Below, we provide an example of one such question.

> **Question 1 (Greek Letter PSI ($\Psi$)):** "Consider a 5x3 grid of points labeled A to O. The points are arranged such that A, B, C form the top row; D, E, F the second row; G, H, I the third; J, K, L the fourth; and M, N, O the bottom row. Consequently, A, D, G, J, M is the left column; B, E, H, K, N is the center; and C, F, I, L, O is the right column. Identify which sets of these locations can form the shape of the uppercase Greek letter PSI ($\Psi$)."

Notably, none of the advanced models successfully solved this problem. For humans, however, the solution becomes relatively straightforward to discern once the problem is visualised as a grid, leveraging our innate spatial reasoning skills. There are two primary sets of solutions to this task, which are presented visually in Figure 2 and detailed systematically in Table 1.

### 4.2 Category 2: Math-based questions

In this category, as its' name implies, the input prompt consists of trivial, yet engaging, mathematical questions where slight modifications in the questions, e.g., slightly tilting the input triangle 3 results in an exponential increase in difficulty.

 

Figure 2: Grid-based visual representation of the solution corresponding to Greek Letter PSI (Ψ) question.

Table 1: The two sets of solutions shown here in a tabular format in response to the Greek Letter PSI (Ψ) question.

| Solution 1 | Solution 2 |
|---|---|
| A-D-G: Vertical straight line. | A-D-G-J: Vertical straight line. |
| G-H-I: Horizontal straight line. | J-K-L: Horizontal straight line. |
| C-F-I: Vertical straight line. | C-F-I-L: Vertical straight line. |
| B-E-H-K-N: Vertical straight Line. | B-E-H-K-N: Vertical straight Line. |

**Question 2 (Area of the Red Triangle):** "Can you compute the area of the red triangle as a function of $d$?"

In Figure 3, we ask the model to determine the area of the red triangle; however, no existing model is currently capable of computing it. The figure is deliberately tilted to increase the complexity of data extraction and contains redundant, non-informative elements. In real-world problem-solving, we should not expect only the relevant information to be available or for it to be neatly aligned. Solving this particular problem requires only a basic understanding of geometry. We have $l_1 = d/2$ because the two angles at the top are both equal to $\varphi$. Similarly, $l_2 = 2d/5$ because $BC$ is parallel to $DE$, $BF$ and $FC$ are each of length $d$, and the angle at point $F$ is $90°$.

## 4.3 Category 3: Logical Puzzles

Below we provide two samples from this category of questions, which, as their name implies, are puzzles that put logical and reasoning skills to the test.

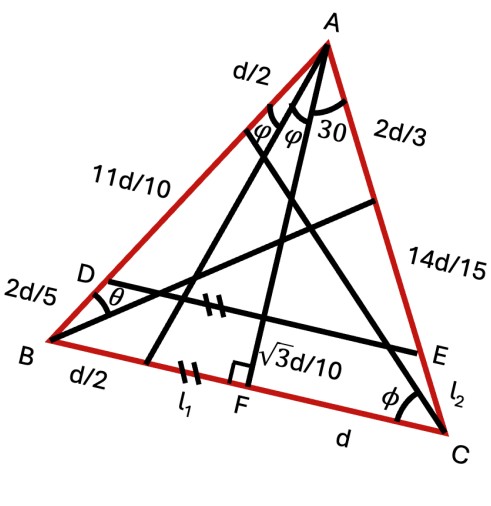

Can you compute the area of the red triangle as a function of d?

Figure 3: Figure showing the image proposed for the prompt asked to compute the area of the red triangle. The prompt is also depicted below the figure in the shaded box.

**Question 3 (Move the Sticks):** "Move three sticks on the left in the provided image (Figure in 4) so that the fish swims to the opposite side." Talwalkar (2019)

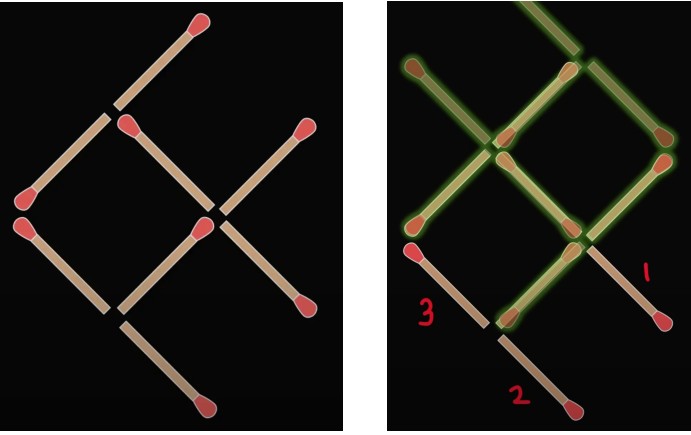

Figure 4: For **Question 3**: The Left figure shows the puzzle, and the right figure shows the answer.

Despite the apparent simplicity of this puzzle for human cognition, our findings indicated a significant challenge for the models; none of the tested systems were able to arrive at the correct solution.

The correct solution, which is visually detailed on the right side of Figure 4, necessitates the strategic repositioning of three specific sticks from the initial configuration. As described, if we identify the sticks that form the fish's head and upper body in the solved state (highlighted in green in Figure 4), the solution involves moving the three lowest sticks as shown in the figure. By relocating these three sticks, the visual representation of the fish is altered to appear as if it is swimming in the reverse direction.

### 4.4 Evaluated LLMs

As part of the submission process, we also asked contributors to report the LLMs with which they had tested their "unreasonable" questions. After consolidating naming variants, the reported models included GPT o1 Pro Mode, DeepSeek-R1, GPT-4o mini, Gemini Advanced 2.0 Flash Thinking, Claude 3.7 with Extended Thinking, Grok 3 Think, Llama 3.1 405B, GPT o1, Gemini 2.0 Flash Thinking, Gemini 2.5 Pro, GPT-4o mini with reasoning, ChatGPT o3, GPT-4o, Claude 3.7 Sonnet, GPT o3, ChatGPT o3 Plus, Gemini 2.5 Pro Preview 03-25, and ChatGPT o4-mini.

At the time of writing, none of these models answered the full set of `Unreasonable Benchmark` questions correctly. Among the evaluated models, Gemini 2.5 Pro achieved the strongest performance, producing correct answers for 21 questions. These results underscored both the rapid pace of progress in the field and the continued value of benchmarks such as `Unreasonable Benchmark` for tracking persistent failure modes.

## 5 Limitations of the `Unreasonable Benchmark`

The Unreasonable Benchmark was built as a collective effort of authors. Given the spontaneous collection nature of the dataset through a crowd-sourcing process, some areas of application might be missing, but will be added as a future addition to the benchmark. Furthermore, we double checked the answers and solutions proposed, but for future works, further refinements and checks could be added to the benchmark. Despite this, we would like to stress that the benchmark is conceptually thought of as an ongoing process, continuously updating itself for questions and models to be tested.

## 6 Ethics Considerations

The `Unreasonable Benchmark` dataset is composed of questions that seek to test an LLM's ability to solve simple yet at times complex problems. The questions were gathered via voluntary contributions from ML researchers and practitioners from both academia and industry. Additional details about the process and how one can contribute to the dataset are found here: https://groups.google.com/g/ml-news/c/HfxR7OIIWrI/m/ysbK-OfdBwAJ?pli=1. It is important to note that the `Unreasonable Benchmark` dataset does not include any personally identifiable information (PII) or other forms of sensitive information. The dataset is purely designed with the goal of providing a new window into how LLM (and the broader AI) benchmarking could evolve, in the process assisting the research community with ques-

tions that could pave the way towards successful AGI development. Similar to prior work, e.g., Cobb et al. (2023), the dataset is released under a Creative Commons Attribution-ShareAlike (CC BY-SA) license and is intended for both academic and research use.

## 7 Conclusion

In this paper, we introduced the `Unreasonable Benchmark`, which consists of 100 questions that no advanced reasoning models can solve today. Current evaluation methods often focus on the limits of model performance but may not adequately assess reliability in routine, low-difficulty scenarios where LLMs surprisingly falter. This new benchmark addresses this gap by systematically testing model performance on basic reasoning and everyday tasks that, despite their apparent simplicity, frequently reveal persistent weaknesses in current LLMs.

The core contribution of this paper is a crowdsourced dataset designed to identify these "negative surprises"—instances where models underperform against reasonable expectations. Unlike static benchmarks, "The Unreasonable Benchmark" evolves by incorporating new failure modes as they are discovered and removing questions once newer models resolve them. This ensures its continued relevance and utility in guiding LLM development. The dataset is made available to inspire developers to tackle these problems, with the ultimate aim of rendering the benchmark's challenges no longer "unreasonable".

By focusing on these unexpected failures, the benchmark serves as a diagnostic tool, highlighting areas where LLMs need targeted improvements to enhance their robustness, address systematic errors, and build greater trust in their ability to handle common tasks accurately. The overarching goal is to drive the development of LLMs that are not only capable of complex feats but are also reliably competent in everyday situations, thereby fostering more resilient and impressive artificial intelligence. The ongoing refinement of this benchmark will continue to pinpoint persistent shortcomings, guiding the community towards building more consistently dependable LLMs. At the same time, the current version of `Unreasonable Benchmark` is subject to a few unavoidable limitations. One important limitation is the relative small dimensionality of the benchmark, which however, is conceptually designed as a living benchmark to be increased in size and variability with time and the contribution of several other authors. A second limitation concerns ambiguity and the subjectivity of some answers. Certain questions may admit multiple plausible interpretations, particularly when instructions are underspecified or depend on implicit assumptions. In such cases, disagreement may arise not only between models and evaluators, but also across human annotators, such that benchmark performance may partly reflect agreement with the intended interpretation rather than purely objective reasoning ability. Moreover, would like to further stress the point, that the benchmark should be interpreted as a diagnostic tool whose utility depends on good governance and continued refresh, not as a one-time definitive leaderboad, allowing for continous update and the intrinsic dynamic nature of it.

## Broader Impact Statement

The Unreasonable Benchmark, a dynamic, crowdsourced dataset is designed to systematically evaluate the reliability of large language models (LLMs) on routine reasoning and

everyday tasks, that are considered simple to be solved for humans, yet remain challenging for current models. Its primary societal impact lies in improving how progress in AI systems is measured, shifting attention from isolated demonstrations of high-level competence toward robustness, consistency, and failure awareness. The possible potential positive impacts are the following. First of all, by highlighting persistent and non-obvious failure modes, the unreasonable benchmark could support the development of more dependable AI systems, particularly in domains where errors in seemingly simple tasks may have disproportionate consequences Kiden et al. (2024). The dataset's evolving nature, moreover, might mitigates overfitting to static benchmarks and encourages continuous diagnostic evaluation rather than leaderboard optimization. As with any benchmark, there is a risk that models may eventually be optimized specifically to perform well on this dataset without addressing broader underlying weaknesses, partially undermining its diagnostic value. Finally, while the dataset avoids sensitive or personal data, its crowdsourced nature may reflect implicit cultural or educational biases in what is considered "reasonable," which could influence conclusions if not continuously diversified and critically examined. The benchmark is explicitly designed as a living artifact, with questions retired once they are no longer informative and new ones added as novel failure modes emerge. This design choice, together with open access and clear framing as a diagnostic—not certification—tool, helps reduce misuse and overinterpretation.

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
