# OpenReview forum: "The Unreasonable Benchmark"
_DMLR — Accepted by DMLR_

### Review · Reviewer_VT9N · 2026-04-16

**Recommendation:** 4
**Confidence:** 3

**Summary Of Contributions:**

The authors introduce the "Unreasonable Benchmark," which serves as a dynamic dataset created through crowdsourcing to assess Large Language Models (LLMs) during their execution of fundamental logical reasoning tasks and common human activities. The dataset tests model performance through negative surprises which occur when advanced models fail to handle simple tasks. The benchmark covers three different types of testable materials which include text and mathematical elements together with logical puzzle components. The dataset functions as an ongoing resource because its creators will remove solved questions from the system to introduce new failure points which need testing. The dataset is publicly hosted on Hugging Face under a CC BY-SA license.

**Strengths:**

The main advantage of this submission lies in its conceptual framework. The authors direct their research to investigate "negative surprises" because this aspect represents a fundamental element of Artificial General Intelligence (AGI) that requires machines to execute basic tasks with complete reliability. The dataset exhibits dynamic characteristics which serve as important research resources because static benchmarks become outdated through data contamination and specific optimization methods.

**Audience:**

Yes

**Claims And Evidence:**

The qualitative claims which state that a dynamic low-difficulty benchmark is essential have strong support from existing research. The claim that advanced models fail to solve these tasks needs more evidence which can be shown through documented proof. The manuscript requires standard evaluation metrics and baseline tables to demonstrate that the 20+ models mentioned achieve success rates of 0% or nearly 0%.

**Datasets And Benchmarks:**

The dataset is hosted on Hugging Face and is available under a CC BY-SA 4.0 license. The authors explain the data collection (crowdsourcing via a Google Group).The authors established their maintenance plan through two methods which depend on community contributions and manual author reviews, but their current system needs a complete curation system to achieve long-term sustainability.

**Extended Submissions:**

N/A

**Limitations:**

The authors adequately acknowledge several limitations because the dataset size is small and the contributor base has geographical and demographic restrictions. The authors correctly identify that models will evolve to perform efficiently on this unique type of trick questions yet their fundamental reasoning abilities will remain unchanged.

**Requested Changes:**

- Reconcile the conflicting numbers regarding the dataset's size (100+ questions vs. 70 questions).
- Provide a comprehensive table detailing the performance of the >20 LLMs mentioned in Section 4.4. Include the methodology for how these models were tested (e.g., zero-shot, few-shot, specific system prompts) to substantiate the claim that "none of these models are capable of successfully answering".
- While the dynamic nature is appreciated, launching with a slightly larger pool of questions (e.g., 200+) would provide a stronger initial foundation.
- Beyond separating by modality (Text, Math, Logic), it would be helpful to categorize why the models fail (e.g., spatial reasoning deficit, distraction by redundant information, pattern-matching over-reliance).

**Strengths And Weaknesses:**

## Strengths:
- The focus on mundane, routine failures addresses a critical blind spot in current LLM evaluation paradigms, which heavily skew toward extreme difficulty.
- The "living benchmark" approach, retiring solved questions and adding new ones, effectively mitigates the risk of models overfitting to the test set aka Goodhart's Law.
- The dataset is fully open-source and clearly licensed.


## Weaknesses:
- The manuscript contains contradictory statements regarding the size of the dataset. Section 1 states it is composed of "more than 100-questions" , while the Conclusion refers to it as consisting of "70 questions".
- Section 4.4 claims over 20 LLMs were evaluated (including Gemini Advanced, Grok 3, DeepSeek-R1, etc.). However, the paper lacks a formalized table or results section detailing the exact accuracy, evaluation protocols, or prompting strategies used to verify that these models consistently failed across the entire dataset.
- Even at 100 questions sourced from ~30 contributors, the dataset is currently quite small to serve as a comprehensive benchmark.

---

### Review · Reviewer_St4c · 2026-04-18

**Recommendation:** 4
**Confidence:** 2

**Summary Of Contributions:**

The authors introduce "The Unreasonable Benchmark," a dynamic and crowdsourced dataset designed to evaluate Large Language Models (LLMs) on routine, low-difficulty tasks where they surprisingly fail. Unlike traditional "frontier" benchmarks that test the limits of expert knowledge, this benchmark focuses on "negative surprises"—mundane reasoning or spatial tasks that humans find trivial but models consistently mishandle.

Key contributions include:

* A dataset of over 100 non-ambiguous questions across text-based, math-based, and logical puzzle categories.

* A dynamic evaluation framework where solved items are retired and new failure modes are added to maintain relevance.

**Strengths:**

See above

**Audience:**

Yes

**Broader Impact Concerns:**

* Cultural Bias: What is considered a "routine" or "mundane" task can vary significantly across different cultures or education systems.

* Optimization Risks: There is a persistent risk that developers may eventually optimize models specifically for these "unreasonable" edge cases without addressing the underlying reasoning gaps.

**Claims And Evidence:**

The primary claim—that LLMs fail on simple, human-trivial tasks—is well-supported by qualitative examples:

* Spatial Reasoning: Models failed to identify the uppercase Greek letter PSI in a simple 5x3 grid.

* Geometry: Models could not compute the area of a "red triangle" when the image was slightly tilted or contained redundant information.

* Logical Puzzles: Models struggled with a simple matchstick puzzle (moving 3 sticks to reverse a fish) that humans solve with ease.

**Datasets And Benchmarks:**

The dataset is available under a Creative Commons Attribution-ShareAlike (CC BY-SA) license. It is provided in a jsonl format with associated image files in a compressed folder.

**Extended Submissions:**

NA.

**Limitations:**

The authors explicitly acknowledge the following:

* Small Sample Size: The initial release is limited to approximately 100 responses.

* Provenance: Geographic and demographic diversity of the contributors is currently narrow.

* Refinement Needs: The spontaneous nature of the collection means some application areas may be missing and require future updates.

**Requested Changes:**

* Formalize Retirement Protocols: Please provide specific criteria for moving a question to the "solved" archive. Is it based on a single SOTA model passing once, or a performance threshold across multiple independent runs?

* Multimodal Expansion Details: The paper mentions text and image modalities. Expanding on how these multimodal inputs are standardized for evaluation would be beneficial.

**Strengths And Weaknesses:**

Strengths

* Novel Diagnostic Philosophy: By focusing on "unreasonable" failures rather than high-level achievement, the benchmark offers higher diagnostic value for assessing true model robustness and progress toward AGI. The dynamic nature of the dataset (retiring solved questions) prevents the common pitfall of models "gaming" the benchmark through static optimization.

Weaknesses

* Scale and Scope: With only ~100 questions and 30 contributors, the dataset is currently small and potentially limited in its geographic or cultural representation.

* Vague Retirement Criteria: While the paper mentions retiring "solved" items, the threshold for what constitutes a "solved" status (e.g., how many models must pass, or with what consistency) could be more rigorously defined.

* Maintenance Dependency: The long-term utility of the benchmark is heavily dependent on sustained community engagement to add new failure modes.

---

### Review · Reviewer_QXSu · 2026-05-13

**Recommendation:** 2
**Confidence:** 2

**Summary Of Contributions:**

The paper introduces The Unreasonable Benchmark, a crowdsourced and dynamic benchmark for testing LLMs on questions that appear simple or reasonable to humans but are often failed by current models. The benchmark includes multiple tasks, mainly grouped into text-based questions, math-based questions, and logical puzzles. The authors argue that this type of benchmark can reveal “negative surprises” that are missed by harder frontier benchmarks, and they propose to update the dataset over time by adding new failures and marking or removing solved items.

The idea is useful and timely. However, the current paper does not yet provide enough evidence or documentation for a benchmark paper. The main claims about model failures, dataset quality, and long-term usefulness are not supported by a reproducible evaluation protocol.

**Strengths:**

The paper targets an important gap in current LLM evaluation: simple failures are often more revealing than impressive performance on very hard benchmarks.

The dynamic benchmark idea is reasonable. If models solve old failure cases, adding new ones and tracking solved items could keep the benchmark useful.

The dataset is publicly released, and the inclusion of multimodal examples is valuable.

The paper is also easy to understand, and the examples make the motivation clear.

**Audience:**

Yes

**Broader Impact Concerns:**

The broader impact is mostly positive if the benchmark is used as a diagnostic tool. It could help expose simple but important model failures.

The main concern is over interpretation. A model doing well or badly on this benchmark should not be treated as evidence of general reliability, safety, or AGI-level competence. The paper says the benchmark is diagnostic, not certification, but the main text still makes broad claims that go beyond the evidence.

There is also a fairness issue. What counts as “reasonable” or “simple” may depend on education, culture, and familiarity with puzzles. The paper should explain how it will handle this.

**Claims And Evidence:**

The evidence is not strong enough for the main claims.

The paper claims that more than 20 LLMs fail on the benchmark, with only Gemini 2.5 Pro solving 21 questions. But this is presented as a paragraph, not as a reproducible experiment. There are no full model outputs, no exact prompts, no scoring rules, and no per-item results.

The example questions are useful illustrations, but they do not prove dataset-level quality. In particular, the geometry example does not give a full final solution in the text, which makes it hard to verify the reference answer.

**Datasets And Benchmarks:**

This is a dataset/benchmark paper, but the dataset documentation is incomplete. The paper should clearly describe the dataset schema, answer format, image handling, scoring process, versioning, update policy, and maintenance plan.

**Extended Submissions:**

The paper does not clearly state whether this is an extended version of earlier work.

**Limitations:**

The paper’s own limitations section is too short. It mentions that the dataset is crowdsourced and geographically limited, but it does not discuss several central issues: contamination from public puzzles, ambiguity in questions, subjective difficulty, possible cultural bias, unstable benchmark versions, and the risk that models may simply overfit to the public examples.

The benchmark also lacks human baseline results. Since the paper repeatedly says the questions are easy or clear for humans, it should show at least a small human evaluation.

**Requested Changes:**

The paper needs major revision before it can be considered ready.

1. The authors should fix the dataset size inconsistency. The paper says “more than 100” questions in the introduction and dataset section, but the conclusion says the benchmark consists of 70 questions. This is a basic but serious problem.

2. The authors should provide a fixed benchmark snapshot for this submission. A dynamic benchmark is fine, but the paper must clearly state which version was evaluated.

3.The evaluation section needs a full results table. For each model, the paper should report the exact model version, date, prompt, settings, number of attempts, and accuracy.

4.The paper should include the scoring protocol. It is not clear what counts as a correct answer for visual puzzles, geometry questions, or free-form reasoning tasks.

5. The answer validation process needs to be explained. The paper only says answers were “double checked,” but does not say by whom, how disagreements were handled, or whether ambiguous questions were removed.

6. The task taxonomy is too coarse. Text-based, math-based, and logical puzzles are not enough to describe the benchmark. The current examples mix spatial reasoning, visual reasoning, geometry, and puzzle solving, which need different evaluation methods.

7. The claims should be toned down. The paper connects the benchmark to AGI, healthcare, and cybersecurity, but the evidence mostly comes from puzzle-style examples. That connection is not yet demonstrated.

**Strengths And Weaknesses:**

The paper has a clear motivation: LLMs should not only solve hard expert tasks, but also handle simple reasoning, spatial, visual, and puzzle-like tasks reliably. This is a real problem, and the benchmark could be useful as a diagnostic collection of failure cases.

The main weakness is that the paper is not yet rigorous enough as a dataset/benchmark submission. The dataset size is inconsistent across the paper, the evaluation of LLMs is not reproducible, the scoring rules are unclear, and the dynamic benchmark design is not properly specified. The examples are interesting, but three examples and a short model list are not enough to establish the benchmark’s reliability. The paper states that many advanced models fail, but it does not show the actual model outputs, prompts, model versions, decoding settings, or per-question results.